# Mosquitocidal Activity of the Methanolic Extract of *Annickia*
*chlorantha* and Its Isolated Compounds against *Culex pipiens*, and Their Impact on the Non-Target Organism Zebrafish, *Danio rerio*

**DOI:** 10.3390/insects13080676

**Published:** 2022-07-27

**Authors:** Tharwat A. Selim, Ibrahim E. Abd-El Rahman, Hesham A. Mahran, Hamza A. M. Adam, Vincent Imieje, Ahmed A. Zaki, Mansour A. E. Bashar, Hossam Hwihy, Abdelaaty Hamed, Ahmed A. Elhenawy, Eman S. Abou-Amra, Samia E. El-Didamony, Ahmed I. Hasaballah

**Affiliations:** 1Zoology and Entomology Department, Faculty of Science, Al-Azhar University, Nasr City, Cairo 11884, Egypt; tharwat3d@azhar.edu.eg (T.A.S.); dr_mb2020682@azhar.edu.eg (M.A.E.B.); hossam.hwihy@azhar.edu.eg (H.H.); samiaeldidamony.sci.g@azhar.edu.eg (S.E.E.-D.); 2Department of Plant Protection, Faculty of Agriculture, Al-Azhar University, Cairo 32897, Egypt; dribrahimelsayed2014@azhar.edu.eg; 3Health Informatics Department, College of Public Health & Tropical Medicine, Jazan University, Jazan 45142, Saudi Arabia; hamahran@jazanu.edu.sa; 4Department of Hygiene, Zoonosis and Epidemiology, Faculty of Veterinary Medicine, Beni-Suef University, Beni-Suef 62511, Egypt; 5Epidemiology Department, College of Public Health & Tropical Medicine, Jazan University, Jazan 45142, Saudi Arabia; drhamza2018@gmail.com; 6Department of Pharmaceutical Chemistry, Faculty of Pharmacy, University of Benin, Benin City 300001, Nigeria; imieje.vimj@gmail.com; 7Pharmacognosy Department, Faculty of Pharmacy, Mansoura University, Mansoura 35516, Egypt; aawad@mans.edu.eg; 8Chemistry Department, Faculty of Science, Al-Azhar University, Nasr City, Cairo 11884, Egypt; abdelaatyhamed.1@azhar.edu.eg (A.H.); elhenawy_sci@hotmail.com (A.A.E.); 9Organic Chemistry Department, Faculty of Science, Al-Azhar University (Girls), Nasr City, Cairo 11751, Egypt; emansadek.59@azhar.edu.eg

**Keywords:** *A. chlorantha*, *C. pipiens*, non-target, bioactive compounds, DFT, docking

## Abstract

**Simple Summary:**

Mosquito-borne diseases lead to serious public health concerns in tropical and sub-tropical countries worldwide. Due to the developed resistance to synthetic chemical insecticides, their effects on the surrounding environment, and economic concerns, natural products are considered promising agents to be used in vector control programs, due to their safety, low cost, and high activity. Therefore, the aim of this study is to evaluate the larvicidal, developmental, and repellent activities of the methanolic extract of *Annickia chlorantha* and its isolated compounds against the mosquito vector, *Culex pipiens*, besides their toxicity to the non-target aquatic organism, the zebrafish (*Danio rerio*). The results highlight the potential of *A.* *chlorantha* extract and its isolated compounds as effective mosquitocidal agents with a very low toxic effect on non-target organisms.

**Abstract:**

In this study, the crude extract and its isolated compounds from the stem bark of *Annickia*
*chlorantha* were tested for their larvicidal, developmental, and repellent activity against the mosquito vector, *Culex pipiens*, besides their toxicity to the non-target aquatic organism, the zebrafish (*Danio rerio*). The acute larvicidal activity of isolated compounds; namely, palmatine, jatrorrhizine, columbamine, *β*-sitosterol, and *Annickia*
*chlorantha* methanolic extract (AC), was observed. Developmentally, the larval duration was significantly prolonged when palmatine and *β*-sitosterol were applied, whereas the pupal duration was significantly prolonged for almost all treatments except palmatine and jatrorrhizine, where it shortened from those in the control. Acetylcholinesterase (AChE) enzyme showed different activity patterns, where it significantly increased in columbamine and *β*-sitosterol, and decreased in (AC), palmatine, and jatrorrhizine treatments, whereas glutathione S-transferase (GST) enzyme was significantly increased when AC methanolic extract/isolated compounds were applied, compared to the control. The adult emergence percentages were significantly decreased in all treatments, whereas tested compounds revealed non-significant (*p* > 0.05) changes in the sex ratio percentages, with a slight female-to-male preference presented in the AC-treated group. Additionally, the tested materials revealed repellence action; interestingly, palmatine and jatrorrhizine recorded higher levels of protection, followed by AC, columbamine, and *β*-sitosterol for 7 consecutive hours compared to the negative and positive control groups. The non-target assay confirms that the tested materials have very low toxic activity compared to the reported toxicity against mosquito larvae. A docking simulation was employed to better understand the interaction of the isolated compounds with the enzymes, AChE and GST. Additionally, DFT calculations revealed that the reported larvicidal activity may be due to the differing electron distributions among tested compounds. Overall, this study highlights the potential of *A.* *chlorantha* extract and its isolated compounds as effective mosquitocidal agents with a very low toxic effect on non-target organisms.

## 1. Introduction

Mosquitoes are considered to be serious threat for human health, since they transmit the causative pathogens of many diseases; *Culex pipiens* (L.) has a wide distribution worldwide and is considered the main vector of Rift Valley Fever virus, lymphatic filariasis, Western Nile Virus, and other pathogens of public health importance. Chemical insecticides have been used for long time. Besides their developed resistance in mosquitoes, synthetic insecticides disturb the environmental balance and cause harmful effects to the non-target habitat. As such, there is a critical need for searching for new products/by-products with larvicidal or insecticidal properties that are less toxic to the non-target organisms [1,2,3]. Our non-target model used here is zebrafish (*Danio rerio*): they are small freshwater fish characterized by their maintenance ease, rapid development, and good acclimatization in laboratory conditions.

Plant products/by-products have become a promising alternative to synthetic chemical insecticides [4,5,6]. Plants contain phytochemicals that belong to different classes, such as steroids, alkaloids, terpenes, and phenolics. These phytochemicals have been investigated by many researchers for their potential insecticidal activities, revealing environmentally safe, cheap, biodegradable, and reliable alternative solutions to chemical strategies used in insect control [7,8]. In addition to their safety for humans and the environment, insecticides from a plant origin are characterized by broad-spectrum activity and relative specificity in their mode of action, which pave the way for such materials to be used in the control of the eggs, larvae, pupae, and adults of medical insects [9].

The discovery of novel compounds with insecticidal/mosquitocidal properties is critically needed to combat the developed resistance rates. Botanicals contain many active phytochemicals with insecticidal properties, and may be considered as alternatives to hazardous synthetic/chemical insecticides [10]. Among detoxification enzymes, acetylcholinesterase (AChE) and glutathione S-transferase (GST) are key enzymes in insect control strategies. AChE catalyzes the hydrolysis of the neurotransmitter (acetylcholine) in the nervous system, which is affected by synthetic insecticides, botanical insecticides, and secondary fungal metabolites [11], whereas GST plays a pivotal role in detoxification and cellular antioxidant defenses against oxidative stress by conjugating reduced glutathione to the electrophilic centers of natural and synthetic exogenous xenobiotics, including insecticides [12].

*Annickia**chlorantha* (Oliv.) Setten and Maas (family, Annonaceae) is known for several medicinal uses. Decoctions, concoctions, and infusions of the stem bark of *A. chlorantha* are used in the traditional health systems of Nigeria, Cameroon, and other West African countries for the treatment of various ailments, such as stomach problems, rickettsia, typhoid fever, infective hepatitis, malaria, and tuberculosis [13]. *A. chlorantha* is locally known as *Awogba*, *Oso pupa* or *Dokita-igbo* (Yoruba), *Osomolu* (Ikale), *Kakerim* (Boki), and *Erenba-vbogo* (Bini). It is widely distributed along the coasts of West and Central Africa, and is very common in the forest regions of Nigeria [13]. Previous phytochemical studies of the stem bark of *A. chlorantha* resulted in the isolation of berberine and protoberberine alkaloids possessing antimalarial [14], antibacterial [15], and trypanosomicidal properties [16]. The current study aims to evaluate the larvicidal, developmental, and repellent activities of the methanolic extract of *Annickia*
*chlorantha* and its isolated compounds against the mosquito vector, *Culex pipiens*, besides their toxicity to the non-target aquatic organism, the zebrafish (*Danio rerio*).

## 2. Materials and Methods

### 2.1. Plant Sample

The plant species used here is *Annickia chlorantha* (Oliv.) Setten and Maas (formerly, *Enantia chlorantha*). The origin of the plant used, collection, identification, deposited voucher of plant specimen, preparation of extract, isolated compounds (Figure 1), and other related information are published elsewhere [17].

### 2.2. Mosquito Colony

The laboratory strain larvae of the mosquito, *Culex pipiens*, were collected from the established colony at the insectary of Medical Entomology, Animal House, Faculty of Science, Al-Azhar University, Cairo, Egypt. It was reared separately in 40-cm-diameter white enamel bowls containing 1000 mL dechlorinated tap water under laboratory conditions of 27 ± 2 °C, 75 ± 5% relative humidity, and a 14–10 h light and dark photoperiod. Larvae were provided with fish food as their diet daily. Emerged adults were supplied with cotton pieces soaked in a 10% sucrose solution. Females were fed on pigeons’ blood for reproduction and development purposes. The deposited eggs were transferred gently to (30 × 30 × 30 cm) wooden cadges supplied with plastic cups containing 500 mL distilled water to allow hatching [18].

### 2.3. Larvicidal Activity

The larvicidal activity of palmatine, jatrorrhizine, columbamine, *β*-sitosterol, and *Annickia*
*chlorantha* extract (AC) were evaluated against the mosquito vector, *C. pipiens*, according to the World Health Organization bioassay for testing mosquito larvicides [19]. Briefly, twenty-five early-third-instar larvae were picked up from the established colony and transferred to 500 mL plastic cups containing 249 mL dechlorinated tap water + 1 mL of tested concentration solubilized in dimethyl sulphoxide (DMSO), which was employed for the negative vehicle control. The larvae were then treated with different concentrations of compounds; namely, palmatine (10, 20, 40, 60, and 80 μg/mL), jatrorrhizine (25, 50, 100, 150, and 200 μg/mL), columbamine (20, 40, 60, 80, and 100 μg/mL), *β*-sitosterol (50, 100, 150, 200, and 250 μg/mL), and AC (50, 100, 200, 300, and 400 μg/mL). The negative control group was tested at the same conditions (25 larvae in 249 mL dechlorinated tap water + 1 mL DMSO). Mortality was recorded 24-h post-treatment. Each concentration was tested in triplicates. More details about the larvicidal activity of tested materials are presented in the Appendix A.

### 2.4. Developmental Durations

From the established colony, 25 newly hatched larvae were treated with the LC_50_ concentration of the AC methanolic extract/isolated compounds, alongside the untreated (control) group. The larval developmental duration (days) of larval instars (L1–L4) was estimated as the duration consumed by the larval instar to reach the next instar [20]. Pupal developmental duration (hours) was estimated as the period between entering the pupal stage and adult emergence [21]. For each treatment, three replicates were tested.

### 2.5. Biochemical Assay

From the established colony, the 1st-instar larvae were treated with the LC_50_ concentration of tested materials, alongside the untreated (control) group. Each concentration was tested in triplicates. Upon the emergence of the 3rd instar, fifty larvae were collected from each treatment and homogenized in distilled water using a Teflon homogenizer dipped in crushed ice for 5 min. The homogenized samples were centrifugated in a refrigerated centrifuge at 6000 r.p.m for 10 min, and the supernatant was used for further biochemical assays. Acetylcholinesterase (AChE) activity was evaluated using acetylcholine bromide as a substrate [22], whereas glutathione S-transferase (GST) activity was evaluated using 1-chloro-2,4-dinitrobenzene as a substrate [23].

### 2.6. Adult Emergence and Sex Ratio

Twenty-five 3rd-instar larvae were picked up from the established colony and gently transferred to separate cadges of standard capacity (30 × 30 × 30 cm). The larvae were treated with the LC_50_ concentration of the AC methanolic extract/isolated compounds, alongside the untreated (control) group. Treatments were observed until adult emergence. Emerged adults were counted, and dead pupae were quantified and excluded to accurately calculate adult emergence following Khazanie [24]. The sex ratio was calculated according to the method of Shetty et al. [25]. The results were calculated as the mean ± standard error (SE) of three replicates.

### 2.7. Repellency Test

The LC_50_ concentrations of the AC methanolic extract/isolated compounds were directly applied on the ventral surface of the pigeon after removal of the abdominal feathers, and left for 10 min, as previously described elsewhere [1]. The pigeons were placed, for 2 h, in cages containing adult *C. pipiens* females starved for 72 h. Distilled water with the same amount of DMSO was used as a negative control, whereas the commercial repellent, DEET 15% (Johnson Wax, Egypt), was applied as a positive control. The treatments were replicated three times in separate cages. Later, the numbers of fed and unfed females were counted and calculated, as described by Abbott [26], as the following: repellency% = (A% − B%/100 − B%) × 100, where (A) is the percentage of unfed females in treatment, and (B) the percentage of unfed females in control. For the repellency assay, the tested animals received care in compliance with the guidelines of the Animal Research Ethics Committee of Al-Azhar University (Egypt).

### 2.8. Toxicity to the Non-Target Organism

The zebrafish (*Danio rerio*) were collected from the established aquaria at the Laboratory of Fish Rearing, Animal House, Zoology Department, Faculty of Science, Al-Azhar University, Cairo, Egypt. The tested animals received care in compliance with the guidelines of the Animal Research Ethics Committee of Al-Azhar University (Egypt). They were acclimatized in circular aquaria with a volume of 1000 mL. Each aquarium contains 10 fish, supported with 24-h artificial aeration. The fish were fed on fish fodder with a suitable pellet size. Triplicate experiments were performed according to Mount [27]. To shed light on the effect of tested materials on our non-target model, thirty adults of healthy zebrafish were exposed to 100, 200, 400, and 800 µg/mL of each tested material over the course of 96 h. The control group was tested under the same conditions (10 fish in each aquarium in triplicate with the same amount of DMSO), and then, mortality was recorded 96-h post-treatment. Toxicity (%) was estimated according to the formula of Deo et al. [28]:$$\;\mathrm{Toxicity}\;\left(\%\right)=\frac{LC50\;of\;target\;vector\;species}{LC50\;of\;non-target\;organisms\times 100}$$

### 2.9. Statistical Analysis

Descriptive statistics, including the mean and SE, were calculated for each treatment. The mean larval mortality data were subjected to *probit* analysis to calculate regression, and LC_50_ and LC_90_ at 95% confidence limits. Analysis of variance, lower and upper confidence limits, and chi-squared values for both tested mosquito and zebrafish mortalities were performed using SPSS (IBM SPSS ver. 25). The Holm–Sidak post hoc test was used for pairwise comparisons. Data are presented as the mean ± SE. The *p*-value was considered significant at <0.05.

### 2.10. Molecular Modeling

#### 2.10.1. Preparation of Small Molecule

The 3D structures for the tested compounds and reference inhibitors (glutathione and difluoromethyl) were optimized using the PM3 semi-empirical Hamiltonian molecular orbital calculation MOPAC16 package, as implemented in the MOE. 2015 package [29].

#### 2.10.2. Selection of Proteins Structures

The docking experiment was carried out for the target active sites of AChE and GST. AChE (PDB ID: 6ARY Cheung et al. [30]) and GST (PDB ID: 1JLV Oakley et al. [31]) proteins were extracted from (https://www.rcsb.org/, accessed on 7 September 2021). The crystal structure of an insecticide-resistant acetylcholinesterase mutant from the malaria vector, *Anopheles gambiae*, and the crystal structure of glutathione S-transferases isozymes from *An. dirus* species were obtained. MOE 2015 was used for correcting errors of the active sites by the structure preparation process in MOE. After the correction, hydrogens were added, and partial charges (Amber12: EHT) were calculated. Energy minimization (AMBER12: EHT, root mean square gradient: 0.100) was performed.

#### 2.10.3. Binding Site Analysis

The binding site of the receptor was identified through the MOE. The Site Finder program uses a geometric approach to calculate putative binding sites in a protein, starting from its tridimensional structure. This method is not based on energy models, but only on alpha spheres, which are a generalization of convex hulls. The prediction of the binding sites, performed by the MOE Site Finder module, confirmed the binding sites defined by the co-crystallized ligands in the holo-forms of the investigated proteins.

#### 2.10.4. MOE Stepwise Docking Method

The crystal structure of the enzymes was obtained. The parameters and charges were assigned with the MMFF94x force field. Alpha-site spheres were generated using the Site Finder module of MOE. The optimized 3D structure of the compounds and the reference inhibitors were subjected to generate different poses of the ligand using the triangular matcher placement method, which generates poses by aligning ligand triplets of atoms on triplets of alpha spheres represented in the receptor site points; a random triplet of alpha sphere centers was used to determine the pose during each iteration. The pose generated was rescored using the London dG. scoring function. The poses generated were refined with the MMFF94x force field; also, the solvation effects were treated. The Born solvation model (GB/VI) was used to calculate the final energy, and the final poses were assigned a score based on the free energy in Kcal/mol.

#### 2.10.5. Computational Methods

The structures of the isolated compounds were drawn in Gauss View 6.0, and the four structures were optimized using DFT-B3LYP/6-31G methods in Gaussian 09 software. The four optimized geometries are at the minimal point on the potential surface, according to no imaginary frequencies.

## 3. Results

The early-third-instar *Culex pipiens* larvae were treated with different concentrations of tested compounds and the *Annickia chlorantha* extract (AC), alongside the control. According to (Table 1), the obtained LC_50_ and LC_90_ confirmed the larvicidal activity of both AC and isolated compounds. Palmatine had the lowest LC_50_, which was 33.392 μg/mL (27.366–40.343), and an LC_90_ of 81.522 μg/mL (63.405–122.239), whereas AC had the highest LC_50_ of 162.630 μg/mL (144.472–182.347), and an LC_90_ of 433.95 μg/mL (365.123–546.16), with no recorded mortality in the control group. According to the LC_50_ and LC_90_, AC and its isolated compounds can be arranged as follows: palmatine, columbamine, jatrorrhizine, *β*-sitosterol, and (AC). The increased chi-squared values for both AC and the isolated compounds indicate the homogeneity of the tested population.

The newly hatched larvae were treated with the LC_50_ concentration of the AC methanolic extract/isolated compounds to estimate its effect on the developmental duration, alongside the control group. There was a significant larval duration increase (*p* < 0.05) for (AC), palmatine, and *β*-sitosterol in the first-instar larvae, whereas in the second, third, and fourth instars, palmatine and *β*-sitosterol significantly (*p* < 0.05) prolonged larval developmental duration compared with the control group (Figure 2). On the other hand, pupal developmental duration was significantly (*p* < 0.05) prolonged in the (AC), columbamine, and *β*-sitosterol, whereas it was significantly (*p* < 0.05) shortened in palmatine and jatrorrhizine treatments compared to the control (Figure 3).

The first instar larvae were treated with the LC_50_ concentration of tested materials, and then, the successfully emerged third-instar larvae were subjected to a biochemical assay to evaluate acetylcholinesterase (AChE) and glutathione S-transferase (GST) activities. The obtained data revealed that the AChE enzyme showed different activity patterns, where it was significantly (*p* < 0.05) increased in columbamine and *β*-sitosterol, and decreased in (AC), palmatine, and jatrorrhizine treatments compared to the control (Figure 4A). On the other hand, the (GST) enzyme was significantly (*p* < 0.05) increased in all tested materials compared to the control group (Figure 4B).

The emergence percentages of adults treated as third-instar larvae with the LC_50_ concentrations of tested materials were significantly (*p* < 0.05) decreased in all treatments compared to the control group. On the other hand, the tested compounds revealed non-significant (*p* > 0.05) changes in the sex ratio percentages, with a slight female-to-male preference presented in the AC-treated group (Table 2).

For the repellency assay, the LC_50_ concentration from each tested material was directly applied on the ventral surface of the pigeon after abdominal feather removal, to allow starved females to blood-feed. The obtained results revealed that all tested materials possess repellence properties, particularly at the beginning of exposure; interestingly, jatrorrhizine and palmatine recorded higher levels of protection, followed by AC, columbamine, and *β*-sitosterol for 7 consecutive hours compared to the negative control, which recorded zero repellency. By comparing the obtained results with the positive control, the tested materials showed significant differentiation (*p* < 0.01) for AC, palmatine, jatrorrhizine, columbamine, and *β*-sitosterol, except exposure to palmatine and jatrorrhizine at the first hour, which showed non-significant variance (*p* > 0.05) compared with the positive control (DEET), which decreased from 96.0 ± 1.0 h at the first hour to 80.0 ± 1.5 h at the seventh hour (Figure 5). Overall, there was decreased repellent action over time from all tested materials.

The non-target organism model used in this study was zebrafish (*Danio rerio*). It was treated with different concentrations of tested materials, alongside the control group, to estimate their potential effects. Based on the obtained results, the LC_50_ for non-target treatments were (805.5 ± 20.1), (1622.3 ± 108.8), (922.3 ± 22.4), (1273.1 ± 50.5), and (1406.6 ± 66.4) µg/mL, whereas the LC_90_ recorded was (1456.6 ± 43.9), (2850.1 ± 212.8), (1504.9 ± 44.4), (1273.1 ± 50.5), and (1406.6 ± 66.4) µg/mL for palmatine, jatrorrhizine, columbamine, *β*-sitosterol, and (AC) extract, respectively, with no mortality in the negative control group (Table 3).

There was a significant difference in the ratio of toxicity values of the tested compounds and (AC) for zebrafish compared to mosquito larvae. Respectively, the concentration values of LC_50_ comparison were: (4.1:95.9), (5.6:94.4), (6.7:93.3), (9.7:90.3), and (11.6:88.4) (folds, percent of change); whereas the concentrations values of LC_90_ comparison were: (5.6:94.4), (8.0:92.0), (7.9:92.1), (12.9:87.1), and (20.1:78.9) (folds, percent of change). These results confirm that the tested materials have very low toxic activity to the non-target organism compared to their reported toxicity against mosquito larvae (Figure 6).

### Binding Efficacy for Molecular Docking

The post-docking results for AChE and GST showed that all docked isolated compounds showed promising binding efficacy: ΔG of about (−4 Kcal/mol.) and (−6 Kcal/mol), respectively (Table 4). The validation of the docking experiment was confirmed by the low RMSD value (1.01 to 1.97). All investigated compounds, when docked into AChE, showed a binding efficacy value near that of the difluoromethyl value as a reference inhibitor. These compounds displayed higher binding potency than glutathione, with ΔG = −5.387 Kcal/mol (original inhibitor for GST). Jatrorrhizine showed the highest LC_50_ against AChE and GST; we cannot explain this finding by ΔG, due to the little variation in these values. Palmatine showed the highest (E._H.B._ = −8.75 and −18.19 Kcal/mol) against both proteins.

The LIGPLOT tool was used to generate the 2D interaction maps. The 3D chemical interaction between the isolated compounds and 6ARY and 1JLV domains has been visualized and represented in Figure 7.

Furthermore, the hydrophobic zone between the AChE and GST domains and the isolated component was examined, and is represented in Figure 8. The red color represents the hydrophobic region in the binding site, whereas the hydrophilic zone is displayed in green zones.

## 4. Discussion

Insecticides from a natural origin may serve as suitable alternatives to chemical insecticides in the future, as they are relatively environmentally safe and inexpensive. This study intended to highlight the role of isolated botanical compounds as an alternative control measure against *Culex pipiens* mosquitoes. *Annickia*
*chlorantha* methanol extract and its isolated compounds have previously demonstrated promising antiprotozoal potential [17]. Herein, we evaluated the mosquitocidal activities of the AC methanolic extract and isolated compounds from the stem bark of *A. chlorantha* against *C. pipiens*, besides their impact on the non-target organism, *Danio rerio*.

For the larvicidal bioassay, the obtained LC_50_ and LC_90_ confirm that (AC) and its isolated compounds are promising potential candidates against the tested mosquito vector, and palmatine was the most potent compound in the different tested biological aspects. However, in the same range of the obtained lethal concentrations, Elumalai et al. [32] demonstrated the potency of the methanol leaf extract of *Gymnema sylvestre* as an effective larvicide against *C. tritaeniorynchus* larvae (LC_50_ = 28.58 ppm); *Azadirachta recorded* an LC_50_ value of 62.5 μg/mL against *Culex pipiens* [33]. Furthermore, Abutaha et al. [34] found similar results (LC_50_ ranged from 42.6 to 85.4 μg/mL) when 69 extracts from ten plant species were evaluated for toxicity against *C. pipiens.* Additionally, the hexane extract of *Ocimum basilicum* leaves showed an LC_50_ of 16.0 (10.9–22.1) and an LC_90_ of 53.0 (34.6–136.8) μg/mL after 24 h of exposure. However, we report here, for the first time, the larvicidal potential of this medicinal plant and its isolated compounds.

The botanical extracts showed various effects on the growth and development of many insect pests, affecting the larval, pupal, and adult stages, and prolonged/shortened their development [35]; it also reduced the survival rates of larvae and pupae, and affected the adult emergence [36]. Many botanical extracts have been reported to have a pronounced effect on the developmental period, growth, and adult emergence [5,37]. Detoxification enzymes, such as acetylcholinesterase and glutathione S-transferase, are known to be key enzymes in insect pest control strategies. Acetylcholinesterase (AChE) protects insects from chemical poisons, whereas glutathione S-transferase (GST) plays an important role in protection mechanisms against oxidative stress. Herein, AChE showed different activity patterns, where it was significantly increased in columbamine and *β*-sitosterol treatments, and decreased in (AC), palmatine, and jatrorrhizine treatments, whereas GST was significantly increased in all tested materials. In similar studies, Al-Solami [38] found that *Lantana camara* extract significantly restricted the AChE activity in larvae, as compared to larvae treated with other plant extracts and/or positive control, and successfully reduced the GST levels.

The adult emergence percentages were significantly decreased in all treatments, whereas the isolated compounds revealed non-significant changes in sex ratio percentages. Similarly, the aqueous leaf extract of *Calotropis procera* inhibited the adult emergence of *C. quinquefasciatus* to 50% at 183.65 ppm [39]. The hexane extract of *Eucalyptus citriodora*, tested at a 10-ppm concentration, failed to emerge *An. Stephensi* adults [40]. Although the typical sex ratio for *C. pipiens* is 1:1, external factors, such as pesticide exposure, can alter the sex ratio, which often results in a male bias [41]. Our data showed the same typical sex ratios even after exposure to different treatments. In contrast, Steinwascher [42] reported a female preference after exposure to curcumin molecules, due to the digestive flow in the female larval gut being greater than in males, resulting in a higher excretion rate.

The tested materials revealed promising repellence action for 7 consecutive hours. Certain other botanicals have previously been investigated for repellent properties against mosquitoes; for example, *Zanthoxylum armatum, Z. alatum*, *A. indica*, and *Curcuma aromatica* have been reported to possess repellent properties against mosquitoes [43]. Rajkumar and Jebanesan [44] reported that *Solanum trilobatum* leaf extract induced repellent activities against *An. stephensi*, with over 100 min of protection. Additionally, Mullai et al. [45] reported that the skin repellent test at 1.0, 2.5, and 5.0 mg/cm^2^ revealed complete protection ranging from 119.17 to 387.83 min against *An. stephensi* with different extracts from *Citrullus vulgaris*. Recently, Junkum et al. [46] found that the hexane extract of *Ligusticum sinense* afforded remarkable repellency against *Aedes aegypti*, *An. minimus*, and *C. quinquefasciatus*, with median protection times of 5.5, 11.5, and 11.25 h, respectively. Overall, our reported repellency investigations here provide the most extended protection time, which prolonged to seven hours from all tested compounds and AC, and this effect was much more pronounced in palmatine and jatrorrhizine.

Treating an aquatic environment with insecticides of a plant origin to control mosquito larvae or other pests may lead to important risks for non-target aquatic organisms [47]. The current results confirm that isolated materials have a very low toxic activity compared to the reported toxicity of mosquito larvae reported here. Similarly, green-fabricated metal nanoparticles failed to show toxicity against different aquatic organisms. For example, *Pergularia*-*daemia*-synthesized AgNPs did not exhibit any evident toxicity against *Poecilia reticulata* fishes after 48 h of exposure [48]. Haldar et al. [49] reported no toxicity of AgNPs produced using *Putranjiva roxburghii* on *P. reticulata* after 48 h of exposure to the LC_50_ of fourth-instar larvae of *An. Stephensi* and *C. quinquefasciatus.* Additionally, most of the LC_50_ values calculated for the non-target organisms have been found to be extremely higher than the lethal concentrations of the targeted pests [50].

### 4.1. Chemical Interaction with AChE and GST Domain Based on Molecular Docking Studies

Acetylcholinesterase’s active site, “AChE” (PDB ID: 1ACJ), contains two principal binding sites, according to its crystallographic structure: the catalytic active site (CAS) and the gorge-connected peripheral anionic site (PAS) [51]. The CAS is made up of amino acids from the esteratic subsite gorge (Ser200, Glu327, and His440), anionic substrate (Trp84, Glu199, and Phe330), and the acyl binding pocket (Phe288 and Phe299), whereas the PAS is made up of Tyr70, Asp72, Tyr121, Trp279, and Phe290 [52].

All compounds interacted with the binding site in the same manner as the reference inhibitor, and formed a strong H-interaction with vital amino acid residues. Jatrorrhizine formed a strong hydrogen interaction, with important Met244 and π–π interaction (Eint = −8.725 Kcal/mol) with Gly601 and Glu359 (E_ele_ = −18.97 Kcal/mol.). The stronger H- and electrostatic interactions compared to other compounds may explain the higher Ec50for of jatrorrhizine. Moreover, columbamine showed an H-interaction with Ala526. *β*-sitosterol interacted with His663 and Arg675 by stack H-bond. Furthermore, jatrorrhizine and columbamine interacted with glutathione S-transferase (GST) by the same important amino acid residue, Tyr113, through H-interaction, whereas palmatine connected with the water molecules which interacted with the Arg66 backbone. Thus, these compounds have remarkable components that explain their inhibition potency of the AChE and GST domains.

### 4.2. DFT Calculation

Frontier molecular orbitals are significant in a variety of chemical and pharmacological activities [53]. The Highest Occupied Molecular Orbital (HOMO) has the priority to provide electrons, whereas the Lowest Unoccupied Molecular Orbital (LUMO) accepts electrons first [54,55]. Thus, studying the frontier orbital may assist in the investigation of insecticidal action. Tested compounds with a significant differential in activity were chosen for DFT comparison. Iteratively solving the self-consistent field equation yielded the optimized geometry corresponding to the minimum of the potential energy surface. Table 5 lists the LUMO and HOMO energies, as well as the HOMO–LUMO (HL) gaps (in eV) of isolated compounds. These orbitals were able to display the binding manner of a biomolecule with a receptor. The FMOs gap was characterized by the chemical reactivity and kinetic stability of the molecule. The molecule with a high EHOMO reflects the strong ability of the molecule for donating electrons, as well as it being easier to lose the electron of valence to biological media; hence, enhancing interactions with a receptor and vice versa. The considerable difference in larvicidal activity may be due to the differing electron distributions among the isolated compounds. When the molecules’ HL gaps were compared, the order was *β*-sitosterol > jatrorrhizine > palmatine > columbamine. The small HOMO–LUMO gap predicts a high chemical reactivity [56]. This showed that the compounds (palmatine and columbamine) would have a reasonably high activity, which corresponded well with our experimental findings. The HOMO and LUMO maps are shown in Figure 9.

## 5. Conclusions

Herein, the crude extract and its isolated compounds from the stem bark of *Annickia*
*chlorantha* were tested for their larvicidal, developmental, and repellent activity against the mosquito vector, *Culex pipiens*, besides their toxicity to the non-target aquatic model, *Danio rerio*. Developmentally, the tested materials induced a prolonged effect for both the larval and pupal durations. The acetylcholinesterase enzyme showed different activity patterns, whereas the glutathione S-transferase enzyme was significantly increased. The adult emergence percentages were significantly decreased, whereas the sex ratio percentages were not affected by the tested materials. The tested materials (in particular, palmatine and jatrorrhizine) revealed potent repellence action. Finally, the tested materials showed a very low toxicity to the non-target model tested here. However, a docking simulation and DFT calculations were employed to better understand the interaction between the isolated compounds and the obtained results. In conclusion, this study highlights the potential of *A. chlorantha* extract and its isolated compounds as safe and effective mosquitocidal agents, with a very low toxic effect on non-target organisms.

## Figures and Tables

**Figure 1 insects-13-00676-f001:**
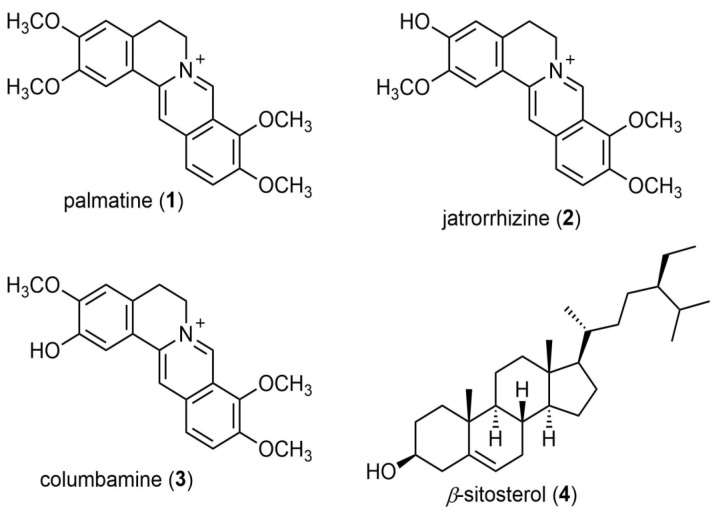
Chemical structures of the isolated compounds.

**Figure 2 insects-13-00676-f002:**
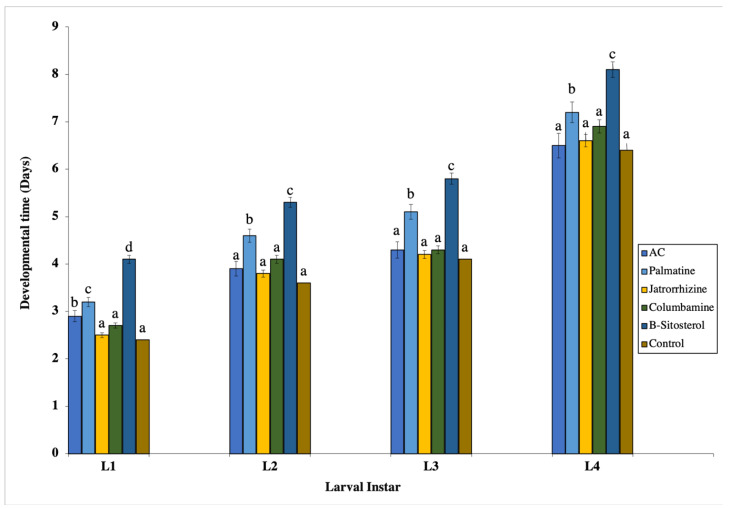
Effect of tested LC_50_ concentrations of palmatine, jatrorrhizine, columbamine, *β-*sitosterol, and *Annickia*
*chlorantha* extract (AC) on the mean developmental durations of *Culex pipiens* larvae. Bars with different letters are significantly (*p* < 0.05) different. Data presented as mean ± SE. Three replicates were used in each treatment. Sample size (*n*) = 75 for the control and L1 groups; *n* = 45 for the L2 group; *n* = 41 for the L3 group; and *n* = 36 for the L4 group. Sample sizes were almost the same for the different tested compounds.

**Figure 3 insects-13-00676-f003:**
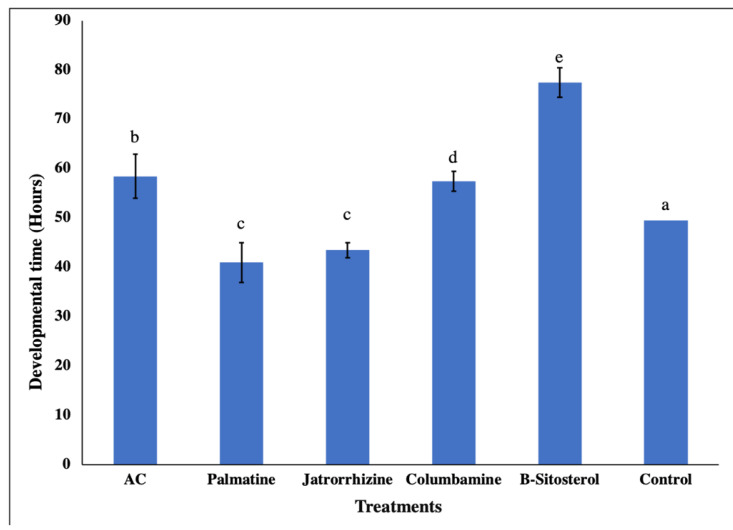
Effect of tested LC_50_ concentrations of palmatine, jatrorrhizine, columbamine, *β*-sitosterol, and *Annickia*
*chlorantha* extract (AC) on the mean pupal developmental durations (hours) of the mosquito, *Culex pipiens*. Bars with different letters are significantly (*p* < 0.05) different. Data presented as mean ± SE of three replicates.

**Figure 4 insects-13-00676-f004:**
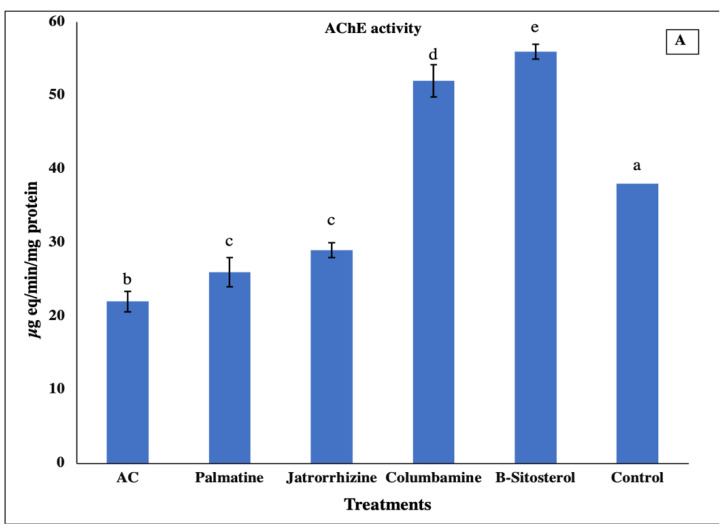
Effect of tested LC_50_ concentrations of palmatine, jatrorrhizine, columbamine, *β-*sitosterol, and *Annickia*
*chlorantha* extract (AC) on the enzymatic activities of acetylcholinesterase (AChE) (**A**) and glutathione S-transferase (GSTs) (**B**) in the third larval instar of the mosquito, *Culex pipiens.* Bars with different letters are significantly (*p* < 0.05) different. Data presented as mean ± SE of three replicates.

**Figure 5 insects-13-00676-f005:**
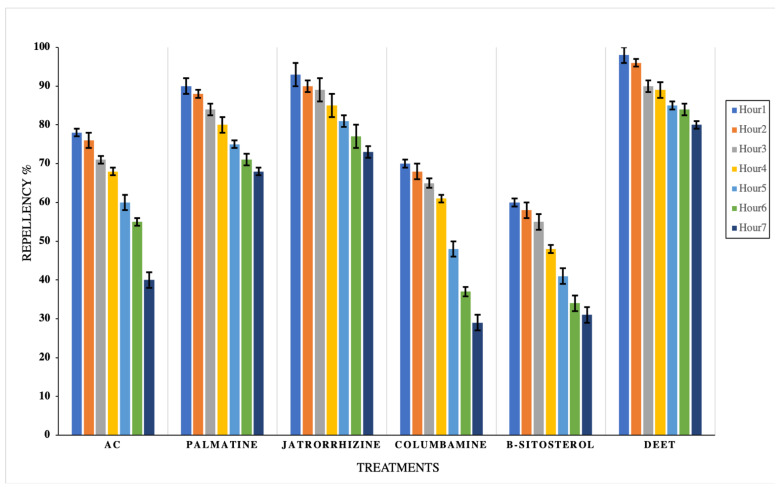
Repellency effect of the LC_50_ concentrations of palmatine, jatrorrhizine, columbamine, *β-*sitosterol, and *Annickia chlorantha* extract (AC) for seven consecutive hours against the mosquito, *Culex pipiens*, adult females. Data presented as average % ± SE. Three replicates were used in each treatment. The negative control group recorded zero repellency action.

**Figure 6 insects-13-00676-f006:**
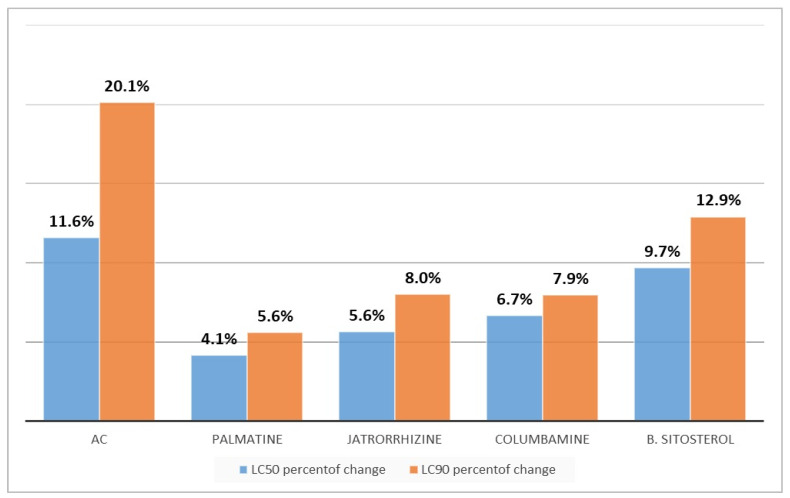
Comparison between lethal concentration values of palmatine, jatrorrhizine, columbamine, *β*-sitosterol, and *Annickia chlorantha* extract (AC) against mosquito larvae and the non-target model.

**Figure 7 insects-13-00676-f007:**
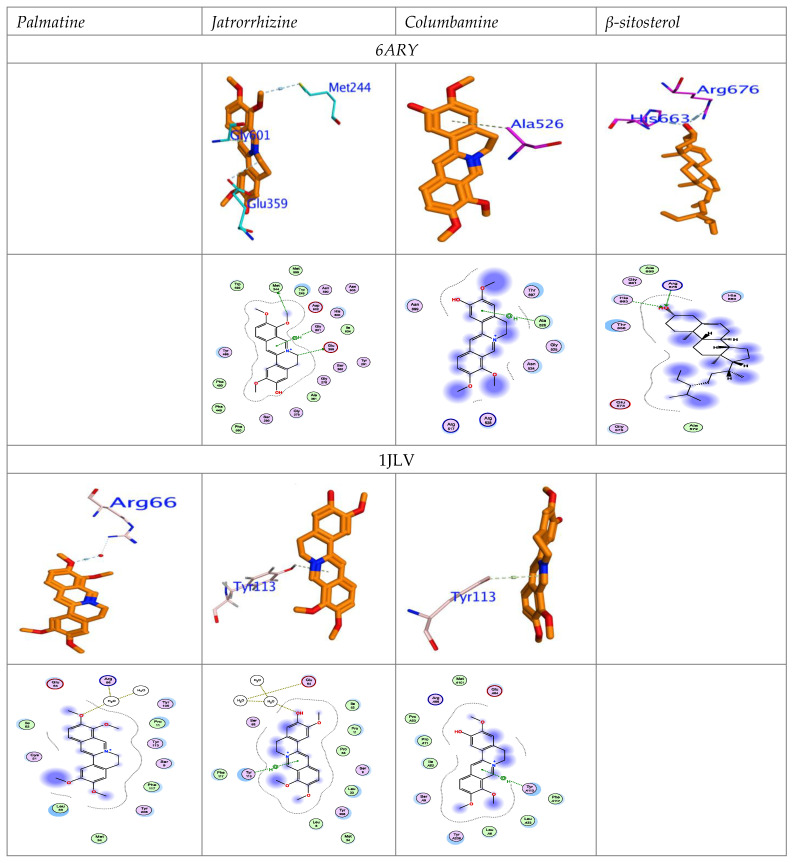
Docked in the active sites of AChE and GST, with corresponding 2D maps. H-Bond is characterized by blue lines, hydrophobic interactions are presented by green dotted lines.

**Figure 8 insects-13-00676-f008:**
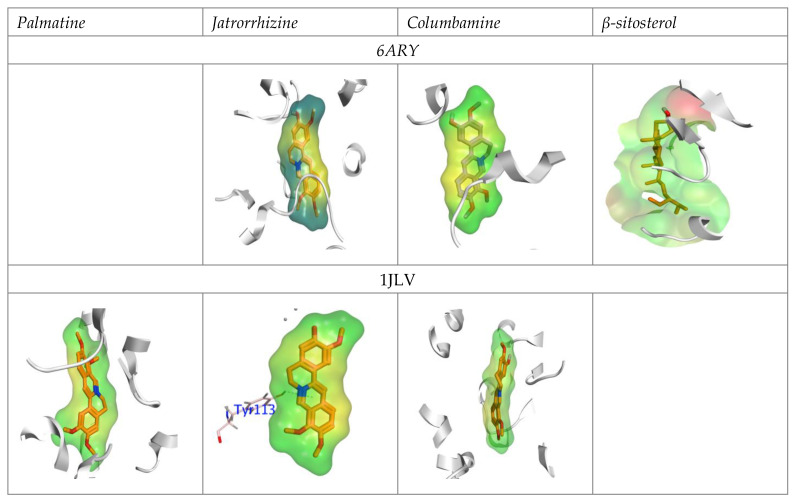
The hydrophobic interaction map for the isolated compounds into the active sites of the AChE and GST domains.

**Figure 9 insects-13-00676-f009:**
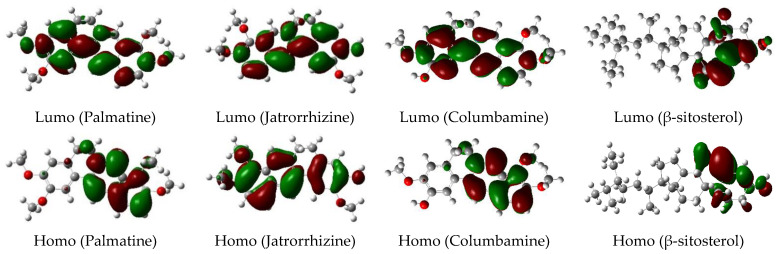
LUMO and HOMO maps for isolated compounds from DFT calculations. The green parts represent positive molecular orbitals, and the red parts represent negative molecular orbitals.

**Table 1 insects-13-00676-t001:** Larvicidal activity of the palmatine, jatrorrhizine, columbamine, *β*-sitosterol, and *Annickia*
*chlorantha* extract (AC) against the third-instar larvae of the mosquito, *Culex pipiens*, 24-h post-treatment.

Compounds (μg/mL)	*A. chlorantha* (AC)	Palmatine	Jatrorrhizine	Columbamine	*β*-Sitosterol
LC_50_ (LCL–UCL) (μg/mL)	162.630 (144.472–182.347)	33.392 (27.366–40.343)	91.343 (74.861–110.929)	61.440 (53.260–70.980)	123.236 (52.365–210.466)
LC_90_ (LCL–UCL) (μg/mL)	433.95 (365.123–546.16)	81.522 (63.405–122.239)	228.135 (174.962–354.875)	119.542 (97.470–171.995)	254.709 (166.189–4711.963)
Regression equation	Y = 0.2499X − 1.0163	Y = 1.3293X − 5.1626	Y = 0.5161X − 6.7236	Y = 1.1267X − 23.867	Y = 0.456X − 14.267
Statistic summary	*d. f.* = 5, *p* *<* 0.001	*d. f.* = 5, *p* < 0.001	*d. f.* = 5, *p* < 0.001	*d. f.* = 5, *p <* 0.001	*d. f.* = 5, *p <* 0.001
χ^2^	19.459	7.121	5.140	6.860	8.006

(LC_50_) concentration that kills 50% of population, (LC_90_) concentration that kills 90% of population, (LCL) lower confidence limit, (UCL) upper confidence limit, (DF) degree of freedom, (χ^2^) chi-squared. Three replicates were used in each treatment, *n* = 375. No mortality was recorded in the negative control group.

**Table 2 insects-13-00676-t002:** Effect of tested LC_50_ concentrations of palmatine, jatrorrhizine, columbamine, *β-*sitosterol, and *Annickia*
*chlorantha* extract (AC) on the adult emergence and sex ratio percentages of the mosquito, *Culex pipiens*.

Treatments	Adult Emergence (%)		Sex Ratio (%) (Mean ± SE)		
	*n*	Mean ± SE	*n*	Males	Females
Control	75	100.0 ± 0.0 a	75	46.67 ± 1.33 a	53.33 ± 1.33 a
*A. chlorantha* (AC)	38	80.13 ± 1.58 b	31	40.72 ± 1.17 b	59.28 ± 1.87 b
Palmatine	35	68.68 ± 2.02 c	24	45.83 ± 4.17 a	54.17 ± 4.17 a
Jatrorrhizine	35	72.34 ± 4.54 d	25	43.52 ± 6.02 a	56.48 ± 6.02 a
Columbamine	36	83.08 ± 1.53 b	30	46.97 ± 1.52 a	53.03 ± 1.5 a
*β*-Sitosterol	36	83.33 ± 4.81 b	30	46.63 ± 1.71 a	53.37 ± 1.71 a
Statistic summary	*d. f.* = 5 *p* < 0.05		*d. f.* = 5 *p* > 0.05		*d. f.* = 5 *p* > 0.05

Data were analyzed by one-way ANOVA, followed by the Holm–Sidak post hoc test, and are presented as the mean ± SE of three replicates. In each column, means followed by different letters differ significantly, *p* < 0.05, *n =* sample size.

**Table 3 insects-13-00676-t003:** Toxicity of the palmatine, jatrorrhizine, columbamine, *β*-sitosterol, and *Annickia*
*chlorantha* extract (AC) to the zebrafish (*Danio rerio*), 96-h post-treatment.

Compounds (μg/mL)	*A. chlorantha*	Palmatine	Jatrorrhizine	Columbamine	*β*-Sitosterol
LC_50_ ± SE (LCL–UCL)	1406.6 ± 66.4 (984.7–4196.9)	805.5 ± 20.1 (641.6–1153.4)	1622.3 ± 108.8 (998.2–16177.5)	922.3 ± 22.4 (743.0–1324.1)	1273.1 ± 50.5 (931.9–2792.6)
LC_90_ ± SE (LCL–UCL)	2157.1 ± 117.6 (1428.7–7232.3)	1456.6 ± 43.9 (1121.4–2270.5)	2850.1 ± 212.8 (1653.3–31863.3)	1504.9 ± 44.4 (1167.7–2348.0)	1974.8 ± 94.0 (1371.7–4830.2)
*d. f.*	4	4	4	4	4
χ^2^	1.93	8.87	3.72	1.87	3.12

See footnote in Table 1.

**Table 4 insects-13-00676-t004:** The docking energy scores (kcal/mol) for the isolated component reference inhibitor.

	ΔG	rmsd	E.vdw	E.Int	E._H.B_	Eele
6ARY						
Palmatine	−4.781	1.991	8.934	−5.667	−7.385	−24.345
Jatrorrhizine	−4.725	1.012	28.914	−14.683	−8.725	−18.975
Columbamine	−4.560	1.731	2.230	−5.482	−7.739	−21.639
β-sitosterol	−4.785	1.450	46.606	−4.659	−5.647	−21.316
Difluoromethyl	−5.01	1.12	33.26	−11.259	−5.78	−28.168
1JLV						
Palmatine	−6.489	1.761	16.228	−17.429	−11.193	−30.506
Jatrorrhizine	−6.256	1.971	6.333	−14.474	−18.195	−25.802
Columbamine	−6.051	1.774	25.000	−11.036	−14.145	−21.342
β-sitosterol	−6.798	1.168	8.083	−16.394	−13.908	−31.302
Glutathione	−5.387	1.89	6.45	−10.67	−11.61	−32.544

ΔG: free binding energy of the ligand from a given conformer, E.Int.: affinity binding energy of hydrogen bond interaction with the receptor, E._H.B._: hydrogen bonding energy between protein and ligand, Eele: electrostatic interaction with the receptor, Evdw: Van der Waals energies between the ligand and the receptor, rmsd: root mean square deviation.

**Table 5 insects-13-00676-t005:** LUMO and HOMO energies and HOMO–LUMO (HL) gaps (in eV) of isolated compounds.

Compound No.	E_HOMO_ (eV)	E_LUMO_ (eV)	HL Gaps (eV)
Palmatine	−0.1074	−0.0352	0.0722
Jatrorrhizine	−0.1891	−0.0362	0.1529
Columbamine	−0.1072	−0.0356	0.0716
β-sitosterol	−0.2279	−0.0279	0.2

## Data Availability

We will provide all data generated in this study upon request.

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
