# Peer review of "Mosquitocidal Activity of the Methanolic Extract of Annickia chlorantha and Its Isolated Compounds against Culex pipiens, and Their Impact on the Non-Target Organism Zebrafish, Danio rerio"

_insects, 2022, doi:10.3390/insects13080676_

Round 1
Reviewer 1 Report
The manuscript reports the mosquitocidal activity of the E. chlorantha methanolic extract and isolated compounds. The methodological design employed biological, biochemical, and molecular docking assays to elucidate the insecticidal potential of the plant species.
First point: it is important to carry out a thorough grammatical and semantic review to adapt the language to avoid deviations (as observed in lines 141, 356, and elsewhere) that could compromise the reading and understanding of the article. Please adapt the paper to the journal's requirements.
Second point: the citations and references are not adequate to the rules of the newspaper. References must be numbered in order of appearance in the text (including table captions and figure legends) and listed individually at the end of the manuscript. Please adapt your citations and references to the journal's requirements.
Third point: How the data were presented in tables and graphs is not uniform from the point of view in which the isolated compounds or methanolic extract are described in the text. This compromises the understanding of the text. Please standardize the presentation of data.
Fourth point: the results are often presented, or parts of the materials and methods, in the discussion and this compromises the argumentation of the data. It is advised to reserve the discussion to contextualize the results. Adjust the data in the results section, as well as methodological aspects in materials and methods.
Sixth point: From the point of view of the biological assay, negative and positive controls were absent or minimally described, compromising the statistical magnitude of the results presented. Insert negative and positive controls into biological assays, as well as positive control into molecular docking.
INTRODUCTION
Lines 65-67: genomic sequencing and the wealth of bioinformatics information make it possible to generalize the information for what? Complement your argument.
Line 79: Enantia chlorantha has had its scientific name redefined and is now accepted as a synonym for the species Annickia chlorantha (Oliv.) Setten & Maas. Update your taxonomic information and indicate the accepted nomenclature for the species, as you can check at http://www.worldfloraonline.org/taxon/wfo-0000667291. See https://doi.org/10.1016/j.jep.2015.10.021 to understand the limitations related to using E. chlorantha as a synonym for A. chlorantha.
In the introduction, there is no information contextualizing the importance of AChE and GST enzymes in insect control. Provide the necessary prior information for the use of these enzymes as a target in the chemical control of C. pipens and contextualize the importance of using these enzymes in biochemical and molecular docking assays.
Materials and methods
The methodology does not contain information about the origin of the plant sample, the deposit in an internationally registered herbarium and the botanical identification code, the methodology for preparing the methanolic extract and fractionation, and the physical methods for structural identification of the phytochemicals used. This information is important to support the phytochemical parameters raised in this study. Has this information been previously published? If yes, provide the reference on the methodology. If not previously published, enter the required data in the text.
Line 107: Didn't the larvicidal test employ a positive control? Positive control is an important parameter to evaluate the effectiveness of phytochemicals with insecticidal action. Please report positive control on materials and methods and results.
Line 110: The indicated reference, WHO (2005), standardized procedures and guidelines for testing mosquito adulticides for indoor residual spraying. The reference does not address the larvicidal activity that is proposed in the Materials and Methods section. Please correct the reference.
Line 120: Is the methodology based on a reference or was it first proposed in this study? If based on another author, include the reference in the text. If it is proposed for the first time in the scientific literature, including the development evaluation parameters and how the data were collected and tabulated to support your results.
Line 122: Is the control positive or negative? Specify control.
Have the repellency (line 148) and toxicity (line 158) assays been approved by your country's animal research and ethics committee or is this approval waived? In any case, provide the information necessary to comply with the journal's Ethical Guidelines for the Use of Animals in Research (mdpi.com/journal/insects/instructions#ethics).
Line 150: Is the repellency test a proprietary methodology or based on an author? What's the reference? Was distilled water the solvent used to solubilize the phytochemical constituents and the methanolic extract? Was the distilled water able to solubilize the isolated compounds and the crude methanolic extract?
Line 166: there is an indication of a control group, but it is not informed whether the control is positive or negative. Describe more emphatically the control employed.
Line 168: finalize the idea and complete the meaning of the sentence here and in other similar points of the methodology.
Line 169: Present the toxicity formula correctly.
Line 170: There is no description of how the LC50 was calculated for the non-target organism. Describe the required information.
Line 174: Include the source of the software used in the statistical analysis and of all other software used in the molecular docking that appears after the statistical analysis.
Line 182: what is the origin of the structures of the AChE and GST proteins? Is there a genetic similarity with C. pipens? Can the results found be extrapolated to C. pipens? Describe this point in the text.
From line 188 onwards: were the free energies of the docked compounds (1, 2, 3, and 4) compared with a positive control to assess the magnitude of coupling with the selected enzymes? Note that this step is an important parameter to determine the insecticidal potential of this study. Please insert a positive control into the molecular docking assay.
RESULTS
It is difficult to read the results when the isolated chemical compounds are presented by number in the text, by name in tables and graphs, or even in a disordered manner. Standardize the presentation of results to make the text easier to read. For exemple, in tables 1, 3, and figure 6 the data were presented in an inverted way, placing EC at the end, different from the other tables and figures. Please standardize the presentation of data
Line 229: the methodology related to the results presented in this step was not informed in the materials and methods section. Please make the necessary adjustments in that session.
Line 270: Are there statistically significant differences between the treatments and the positive control? What statistical data support the result presented? Inform all data, not just the p-value in this item, and all the text when using this tool.
DISCUSSION
The authors did not elucidate the relationship between the insect developmental stages and the selected proteins (AChE and GST), making it difficult to contextualize the results in the discussion. This aspect weakens the methodological path chosen. Please construct meanings between the bioassays that assess C. pipens development, the biochemical assays, molecular docking, and phytochemical/extract constituents in the paper's discussion.
Lines 329-341: there is no semantic link in the paragraph, apparently the sentences have coalesced in the text. Please build a link between line 334 to ensure understanding of the text.
Lines 334-341 just repeat the data found in the results, the scientific community expects an explanation that justifies the variation between the results found. Why did compounds 3 and 4 behave differently from compounds 1 and 2? From the point of view of molecular docking of compounds 1, 2, 3, and 4, why is the free energy of binding similar? How do biochemical assay results relate to modeling for AChE? From this same perspective, how does the GTS behave? Bringing answers to the discussion is important to support the results.
Lines 400-403: Figures 7 and 8 look much more like results than discussions. It is advisable to present the results in the appropriate item. The same happens between lines 389-391, which describe steps more related to materials and methods than discussion. These incompatibilities are noted elsewhere in the text. Please make the text easier to read and understand by rearranging these and other paragraphs in the appropriate sections.
The interaction of compound 2 with Gly501 (Line 394) and the interaction of compound 4 with Arg 575 (line 395) are not found in Figure 7. Does this represent a typo or data collection error?
Again the compounds appear in random order in figures 7 and 8. Organize the data for better understanding.
Table 5 presents results that should not be included in the discussion.
Line 423: From the point of view of larvicidal activity and the DFT calculation, how do these results explain the insecticidal potential of compounds 1, 2, 3, and 4? This discussion needs to support the results found.
Author Response
Response to reviewer (1) comments
First of all, we really appreciate the reviewer’s valuable comments that shaped our manuscript better and added true value to the presented work, as well as the handling editor for his kind consideration of this article for possible publication.
Comments.
- The manuscript reports the mosquitocidal activity of the chloranthamethanolic extract and isolated compounds. The methodological design employed biological, biochemical, and molecular docking assays to elucidate the insecticidal potential of the plant species.
First point: it is important to carry out a thorough grammatical and semantic review to adapt the language to avoid deviations (as observed in lines 141, 356, and elsewhere) that could compromise the reading and understanding of the article. Please adapt the paper to the journal's requirements.
Response
Thanks a lot for these valuable comments. We have revised the MS grammatically and also got some advice from a native English speaker. We hope the revised version meets the journal's requirements.
- Second point: the citations and references are not adequate to the rules of the newspaper. References must be numbered in order of appearance in the text (including table captions and figure legends) and listed individually at the end of the manuscript. Please adapt your citations and references to the journal's requirements.
Response
Thanks for this comment, we have updated the citations and references as per the journal's requirements.
- Third point: How the data were presented in tables and graphs is not uniform from the point of view in which the isolated compounds or methanolic extract are described in the text. This compromises the understanding of the text. Please standardize the presentation of data.
Response
Dear respected reviewer, we are totally agreeing with your point of view, data presentation is now standardized as per your request.
- Fourth point: the results are often presented, or parts of the materials and methods, in the discussion and this compromises the argumentation of the data. It is advised to reserve the discussion to contextualize the results. Adjust the data in the results section, as well as methodological aspects in materials and methods.
Response
Thank you for this valuable comment, we have moved any parts of M&M or Results from the discussion section to their proper places.
- Sixth point: From the point of view of the biological assay, negative and positive controls were absent or minimally described, compromising the statistical magnitude of the results presented. Insert negative and positive controls into biological assays, as well as positive control into molecular docking.
Response
All respect to your valuable comment, unfortunately, we didn’t employ a positive control in the larvicidal assay, which is considered a critical limitation in this study. However, we have illustrated in M&M and in the tables’ footnote that negative controls were used in the presented results. Additionally, for molecular docking we have added the reference inhibitors for AChE and GST in the experimental part in the “Preparation of Small Molecule” subsection.
- Introduction, lines 65-67: genomic sequencing and the wealth of bioinformatics information make it possible to generalize the information for what? Complement your argument.
Response
Thanks for this valuable comment, we have deleted this sentence since we didn't carry out genetic or metabolic bioassays on the non-target model.
- Line 79: Enantia chloranthahas had its scientific name redefined and is now accepted as a synonym for the species Annickia chlorantha (Oliv.) Setten & Maas. Update your taxonomic information and indicate the accepted nomenclature for the species, as you can check at http://www.worldfloraonline.org/taxon/wfo-0000667291. See https://doi.org/10.1016/j.jep.2015.10.021 to understand the limitations related to using chlorantha as a synonym for A. chlorantha.
Response
Thanks for this comment, we have updated the scientific name for (Enantia chlorantha) with Annickia chlorantha (Oliv.) Setten & Maas in the main text and in supplementary file. Thanks again.
- In the introduction, there is no information contextualizing the importance of AChE and GST enzymes in insect control. Provide the necessary prior information for the use of these enzymes as a target in the chemical control of pipens and contextualize the importance of using these enzymes in biochemical and molecular docking assays.
Response
Dear respected reviewer, we have added a paragraph to the introduction to illustrate the importance of selected enzymes and to give a background on the interaction of the tested compounds with AChE and GST. We wrote the following:
“Discovery of novel compounds with insecticidal/mosquitocidal properties is critically needed to combat the developed resistance rates. Botanicals contain many active phytochemicals with insecticidal properties that may consider alternatives to hazardous synthetic/chemical insecticides ]10[. Among detoxification enzymes, Acetylcholinesterase (AChE) and Glutathione S-transferase (GST) are key enzymes in insect control strategies.AChE is catalyzing the hydrolysis of the neurotransmitter (acetylcholine) in the nervous system which affected by synthetic insecticides, botanical insecticides, and secondary fungal metabolites ]11[. While, GST plays a pivotal role in detoxification and cellular antioxidant defenses against oxidative stress by conjugating reduced glutathione to the electrophilic centers of natural and synthetic exogenous xenobiotics, including insecticides ]12[.”
- The methodology does not contain information about the origin of the plant sample, the deposit in an internationally registered herbarium and the botanical identification code, the methodology for preparing the methanolic extract and fractionation, and the physical methods for structural identification of the phytochemicals used. This information is important to support the phytochemical parameters raised in this study. Has this information been previously published? If yes, provide the reference on the methodology. If not previously published, enter the required data in the text.
Response
Thanks for your valuable comments, we have added the following sentence to the M&M section:
“Plant sample”
“Plant species used here is Annickia chlorantha (Oliv.) Setten & Maas (formerly; Enantia chlorantha). Origin of the plant used, collection, identification, deposited voucher of plant specimen, preparation of extract and isolation of compounds and other related information are published elsewhere ]17[.”
- Line 107: Didn't the larvicidal test employ a positive control? Positive control is an important parameter to evaluate the effectiveness of phytochemicals with insecticidal action. Please report positive control on materials and methods and results.
Response
Unfortunately, we didn’t employ a positive control in the larvicidal test which is considered a critical limitation in this study.
- Line 110: The indicated reference, WHO (2005), standardized procedures and guidelines for testing mosquito adulticides for indoor residual spraying. The reference does not address the larvicidal activity that is proposed in the Materials and Methods section. Please correct the reference.
Response
Thank you for this valuable comment, we have replaced this reference with “World Health Organization. (2005). Guidelines for laboratory and field testing of mosquito larvicides. World Health Organization. https://apps.who.int/iris/handle/10665/69101”
Thanks again.
- Line 120: Is the methodology based on a reference or was it first proposed in this study? If based on another author, include the reference in the text. If it is proposed for the first time in the scientific literature, including the development evaluation parameters and how the data were collected and tabulated to support your results.
Response
Thank you for these valuable comments, we have inserted a reference to the developmental durations subsection in the methods.
- Line 122: Is the control positive or negative? Specify control.
Response
Thank you for this valuable comment, as we mentioned before we have illustrated in M&M and in the tables’ footnote that negative controls were used in the presented results.
- Have the repellency (line 148) and toxicity (line 158) assays been approved by your country's animal research and ethics committee or is this approval waived? In any case, provide the information necessary to comply with the journal's Ethical Guidelines for the Use of Animals in Research (mdpi.com/journal/insects/instructions#ethics).
Response
Thank you for this valuable comment, since we wrote the following in these two subsections:
“Tested animals received care in compliance with the guidelines of the animal research ethics committee of Al-Azhar University (Egypt).”
- Line 150: Is the repellency test a proprietary methodology or based on an author? What's the reference? Was distilled water the solvent used to solubilize the phytochemical constituents and the methanolic extract? Was the distilled water able to solubilize the isolated compounds and the crude methanolic extract?
Response
Thank you for your comments, we have inserted a reference to the repellency test subsection in the methods. Additionally, isolated compounds and (AC) methanolic extract were solubilized in dimethyl sulphoxide (DMSO) as mentioned in the larvicidal activity subsection.
- Line 166: there is an indication of a control group, but it is not informed whether the control is positive or negative. Describe more emphatically the control employed.
Response
Thank you for your comments, we have explained this comment before.
- Line 168: finalize the idea and complete the meaning of the sentence here and in other similar points of the methodology.
Response
Thank you for your comment, we wrote the following sentence in the methods:
“The control group was tested under the same conditions (10 fish in each aquarium in triplicate with the same amount of DMSO) then mortality was recorded 96 h post-treatment.”
- Line 169: Present the toxicity formula correctly.
Response
Thank you for your comment, we have edited the formula as per your request.
- Line 170: There is no description of how the LC50 was calculated for the non-target organism. Describe the required information.
Response
Thank you for your comment, we have inserted the following sentence into the statistical analysis subsection “Analysis of variance, lower and upper confidence limits and Chi-square values for both tested mosquito and zebrafish mortalities were done using SPSS (IBM SPSS ver. 25).”
- Line 174: Include the source of the software used in the statistical analysis and of all other software used in the molecular docking that appears after the statistical analysis.
Response
Thank you for your comment, we have inserted the source to the statistical analysis subsection “….. using SPSS (IBM SPSS ver. 25).” Additionally, all the software used in molecular docking has been mentioned in the molecular modeling part.
- Line 182: what is the origin of the structures of the AChE and GST proteins? Is there a genetic similarity with pipens? Can the results found be extrapolated to C. pipens? Describe this point in the text.
Response
Thank you for raising these points, the proteins were extracted from (https://www.rcsb.org/). Crystal structure of an insecticide-resistant acetylcholinesterase mutant from the malaria vector Anopheles gambiae. The crystal structure glutathione S-transferases isozymes from Anopheles dirus species and mentioned in the manuscript at “Selection of proteins structures” subsection.
- From line 188 onwards: were the free energies of the docked compounds (1, 2, 3, and 4) compared with a positive control to assess the magnitude of coupling with the selected enzymes? Note that this step is an important parameter to determine the insecticidal potential of this study. Please insert a positive control into the molecular docking assay.
Response
Thank you for your comment, we have calculated the molecular docking energies for glutathione and difluoromethyl and inserted the values in table 4.
- It is difficult to read the results when the isolated chemical compounds are presented by number in the text, by name in tables and graphs, or even in a disordered manner. Standardize the presentation of results to make the text easier to read. For exemple, in tables 1, 3, and figure 6 the data were presented in an inverted way, placing EC at the end, different from the other tables and figures. Please standardize the presentation of data.
Response
Thank you for these valuable comments, we have replaced the presented numbers for isolated compounds by names in the main text and in tables and figures and kept the uniformity of presented results. Thanks again.
- Line 229: the methodology related to the results presented in this step was not informed in the materials and methods section. Please make the necessary adjustments in that session.
Response
Thank you for your comment, but the pupal development assay determination was previously described in the methods as the following:
“Pupal developmental duration (hours) was estimated as the period between entering the pupal stage till adult emergence ]21[. For each treatment, three replicates were tested.”
- Line 270: Are there statistically significant differences between the treatments and the positive control? What statistical data support the result presented? Inform all data, not just the p-value in this item, and all the text when using this tool.
Response
Dear respected reviewer, for the repellency assay we have rerun a two-way ANOVA using Holm-Sidak method. Statistical summary revealed that multiple comparisons of tested materials versus the positive control (DEET) showed a significance level of P< 0.05. There was significant differentiation (P<0.01) for AC, Palmatine, Jatrorrhizine, Columbamine and β-Sitosterol except exposure to palmatine and jatrorrhizine at the first hour which showed non-significant variance (P> 0.05) if compared with the positive control (DEET). Overall, there was decreased repellent action by time of all tested materials. We wrote in the results section the following:
“For repellency assay, the LC50 concentration from each tested material was directly applied on the ventral surface of the pigeon after abdominal feathers removal to allow starved females to blood feed. Obtained results revealed that all tested materials possess repellence properties, particularly at the beginning of exposure, interestingly jatrorrhizine and palmatine recorded higher levels of protection followed by AC, columbamine and β-sitosterol for 7 consecutive hours compared to the negative control that recorded zero repellency. By comparing the obtained results with the positive control, tested materials showed significant differentiation (P< 0.01) for AC, palmatine, jatrorrhizine, columbamine and β-sitosterol except exposure to palmatine and jatrorrhizine at the first hour which showed non-significant variance (P> 0.05) if compared with the positive control (DEET) that decreased from 96.0 ± 1.0 h at the 1st hour to 80.0 ± 1.5 h at the 7th hour (Figure 5). Overall, there was decreased repellent action by time of all tested materials.”
- The authors did not elucidate the relationship between the insect developmental stages and the selected proteins (AChE and GST), making it difficult to contextualize the results in the discussion. This aspect weakens the methodological path chosen. Please construct meanings between the bioassays that assess pipens development, the biochemical assays, molecular docking, and phytochemical/extract constituents in the paper's discussion.
Response
Dear respected reviewer, thank you for raising these points to enhance the manuscript, we have mentioned the relation between our finding and molecular docking in the revised manuscript, we wrote the following:
“Jatrorrhizine formed strong hydrogen interaction with important Met244 and p-p interaction (Eint = -8.725 Kcal/mol) with Gly601 & Glu359 (Eele = -18.97 Kcal/mol.). The stronger H- & electrostatic interactions than other compounds may explain the higher Ec50for of jatrorrhizine.”
- Lines 329-341: there is no semantic link in the paragraph, apparently the sentences have coalesced in the text. Please build a link between line 334 to ensure understanding of the text.
Response
Dear respected reviewer, we do agree with your comment, we have inserted a linking sentence to better understand this paragraph, we wrote in the discussion “Detoxification enzymes such as Acetylcholinesterase and Glutathione S-transferase are key enzymes in insect pest control strategies”
- Lines 334-341 just repeat the data found in the results, the scientific community expects an explanation that justifies the variation between the results found. Why did compounds 3 and 4 behave differently from compounds 1 and 2? From the point of view of molecular docking of compounds 1, 2, 3, and 4, why is the free energy of binding similar? How do biochemical assay results relate to modeling for AChE? From this same perspective, how does the GTS behave? Bringing answers to the discussion is important to support the results.
Response
Dear respected reviewer, the free energy of tested compounds is very close but not identical while other parameters as E.Int. (affinity binding energy of hydrogen bond interaction with the receptor), E.H.B. (hydrogen bonding energy between protein and ligand), Eele (electrostatic interaction with the receptor), Evdw (Van der Waals energies between the ligand and the receptor), and rmsd (root mean square deviation) are different which cause a different interaction with AChE and GST. We added these sentences to the main text:
“All investigated compounds when docked into AChE showed binding efficacy value near to that of difluoromethyl value as a reference inhibitor. These compounds displayed higher binding potency than glutathione with ΔG = -5.387 Kcal/mol (original inhibitor for GST). Jatrorrhizine showed the highest LC50 against AChE & GST, we cannot explain this finding by ΔG due to the little variation in these values. Compound 1 showed the highest (E.H.B. = -8.75 & -18.19 Kcal/mol) against both proteins.”
- Lines 400-403: Figures 7 and 8 look much more like results than discussions. It is advisable to present the results in the appropriate item. The same happens between lines 389-391, which describe steps more related to materials and methods than discussion. These incompatibilities are noted elsewhere in the text. Please make the text easier to read and understand by rearranging these and other paragraphs in the appropriate sections.
Response
Dear respected reviewer, we do agree with you, figure 7 and 8 are related to the results section than the discussion section but it seems to be moved by due to the journal’s format, however, we have shifted these lines and their figures to the results section as per your request.
- The interaction of compound 2 with Gly501 (Line 394) and the interaction of compound 4 with Arg 575 (line 395) are not found in Figure 7. Does this represent a typo or data collection error?
Response
Thanks for your comment. We have corrected it in the text.
- Again the compounds appear in random order in figures 7 and 8. Organize the data for better understanding.
Response
Dear respected reviewer, we have reorganized the data in all tables and figures as per your request.
- Table 5 presents results that should not be included in the discussion.
Response
Dear respected reviewer, we do agree with you here too, we have shifted it to the results section as per your request.
- Line 423: From the point of view of larvicidal activity and the DFT calculation, how do these results explain the insecticidal potential of compounds 1, 2, 3, and 4? This discussion needs to support the results found.
Response
Dear respected reviewer, thanks for this valuable comment, we have added an explanation in the text, we wrote the following:
“These orbitals were able to display the binding manner of a biomolecule with a receptor. FMOs gap was characterized by the chemical reactivity and kinetic stability of the molecule. The molecule with high EHOMO reflects the strong ability of the molecule for donating electron as well as easier for losing electron of valence to biological media, and hence enhancing interactions with a receptor, and vice versa.”

Reviewer 2 Report
This manuscript entitled “Mosquitocidal activity of the methanolic extract of Enantia chlorantha and its isolated compounds against Culex pipiens and their impact on the non-target organism zebrafish, Danio rerio.” describes a study of several chemical compounds that may serve as a novel mosquito repellent. The authors first identified four chemical compounds extracted from EC plants. Using cultured mosquitos, larvicidal, developmental and repellent activity against mosquito were tested. They also checked the toxicity in zebra fish and found no or very low toxic affection on non-target model. Finally the authors performed docking stimulation to mimic interactions between AchE/GST and the chemicals from EC.
Overall, this paper combines a series of genetics, behavior, biochemistry and structure techniques. Most of the data from this paper are high quality and supportive to the conclusions. This paper is also well written and easy to follow. Mosquito is one of the main vectors of a lot of tropical diseases. Developing new natural derived repellents like this paper is crucial for public health. I have the following comments/suggestions to improve the manuscript prior to publication.
1. The reason for testing AchE/GSTs enzyme activity is not clear to me when read the text. Why these two enzymes not others are picked? Is there any solid evidence that AchE/GSTs are the main targets for these chemical compounds extracted from EC?
2. Figure 2, it’s difficult to corelate the figure legends to their bars in the figure, either use different colors or list their name under each bar.
3. I might overlook the reason, but there is no structure data for the interaction between Palmatine and AchE (6ARY), same for b-sitoterol with GST(1JLV) in the figure 7 and 8, please explain the reason if not in the paper.
4. Line 301: explain what is RMSD short for?
5. From Figure 5, Palmatine and Jatrorrhizine have best repellency compared to other chemicals, can the structure study in the following parts explain this phenomenon? If not, what is the potential mechanism?
Author Response
Response to reviewer (2) comments
First of all, we really appreciate the reviewer’s valuable comments that shaped our manuscript better and added true value to the presented results, as well as the handling editor for his kind consideration of this article for possible publication.
Comments.
- This manuscript entitled “Mosquitocidal activity of the methanolic extract of Enantia chlorantha and its isolated compounds against Culex pipiens and their impact on the non-target organism zebrafish, Danio rerio.” describes a study of several chemical compounds that may serve as a novel mosquito repellent. The authors first identified four chemical compounds extracted from EC plants. Using cultured mosquitos, larvicidal, developmental and repellent activity against mosquito were tested. They also checked the toxicity in zebra fish and found no or very low toxic affection on non-target model. Finally the authors performed docking stimulation to mimic interactions between AchE/GST and the chemicals from EC. Overall, this paper combines a series of genetics, behavior, biochemistry and structure techniques. Most of the data from this paper are high quality and supportive to the conclusions. This paper is also well written and easy to follow. Mosquito is one of the main vectors of a lot of tropical diseases. Developing new natural derived repellents like this paper is crucial for public health.
Response
Thanks a lot for these positive comments, really appreciate.
- The reason for testing AchE/GSTs enzyme activity is not clear to me when read the text. Why these two enzymes not others are picked? Is there any solid evidence that AchE/GSTs are the main targets for these chemical compounds extracted from EC?
Response
Thanks a lot for your valuable comment, we have inserted the following paragraph into the intro section to increase readability and better understand this question:
“Discovery of novel compounds with insecticidal/mosquitocidal properties is critically needed to combat the developed resistance rates. Botanicals contain many active phytochemicals with insecticidal properties that may consider alternatives to hazardous synthetic/chemical insecticides ]10[. Among detoxification enzymes, Acetylcholinesterase (AChE) and Glutathione S-transferase (GST) are key enzymes in insect control strategies.AChE is catalyzing the hydrolysis of the neurotransmitter (acetylcholine) in the nervous system which affected by synthetic insecticides, botanical insecticides, and secondary fungal metabolites ]11[. While, GST plays a pivotal role in detoxification and cellular antioxidant defenses against oxidative stress by conjugating reduced glutathione to the electrophilic centers of natural and synthetic exogenous xenobiotics, including insecticides ]12[.”
- Figure 2, it’s difficult to correlate the figure legends to their bars in the figure, either use different colors or list their name under each bar.
Response
Thanks a lot for your valuable comment, we have changed the way of all presented figures to make them easier to read, thanks again.
- I might overlook the reason, but there is no structure data for the interaction between Palmatine and AchE (6ARY), same for β-sitoterol with GST(1JLV) in the figure 7 and 8, please explain the reason if not in the paper.
Response
Thanks a lot for your valuable comment, we really value your feedback. The missing interactions between Palmatine and AchE (6ARY) and β-sitosterol with GST(1JLV) were added in figures 7 and 8.
- Line 301: explain what is RMSD short for?
Response
Thanks a lot for your valuable comment, we have added the explanation in the footer of table 4.
- From Figure 5, Palmatine and Jatrorrhizine have best repellency compared to other chemicals, can the structure study in the following parts explain this phenomenon? If not, what is the potential mechanism?
Response
Thanks a lot for your valuable comment. Palmatine, jatrorrhizine and columbamine compounds are very close in structure, so it might seem strange for them to have such differences in their behavior in different assays, but according to a huge number of previous studies, a slight difference in the structure, even a difference in the position of a substituent, could enhance or demolish the activity. The structural calculations in this study, DFT calculations, and molecular docking pointed out how the isolated compounds interact and bind to AChE and GST enzymes only. For the repellency test, those compounds which show a high effect should interrupt the insect’s olfactory responses. Knowing that olfactory sensory neurons of the yellow fever mosquito, Aedes aegypti, can specifically recognize one enantiomer of the host attractant, 1-octen-3-ol, and responds with much lower sensitivity to structurally similar compounds (Bohbot and Dick-ens, 2009) could explain how C. pipens responds more to Palmatine and Jatrorrhizine rather than Columbamine and β-Sitosterol. Anyway, this is an interesting point of research raised here and we will be keen to deeply investigate it in the future. Thanks again.

Round 2
Reviewer 1 Report
The adaptations of the manuscript are in agreement with the review. Thus, it meets the requirement for publication in the journal.